# Sources and Characteristics of Polycyclic Aromatic Hydrocarbons in Ambient Total Suspended Particles in Ulaanbaatar City, Mongolia

**DOI:** 10.3390/ijerph16030442

**Published:** 2019-02-02

**Authors:** Batdelger Byambaa, Lu Yang, Atsushi Matsuki, Edward G. Nagato, Khongor Gankhuyag, Byambatseren Chuluunpurev, Lkhagvajargal Banzragch, Sonomdagva Chonokhuu, Ning Tang, Kazuichi Hayakawa

**Affiliations:** 1Graduate School of Natural Science and Technology, Kanazawa University, Kanazawa 920-1192, Japan; batdelger@stu.kanazawa-u.ac.jp; 2Department of Environment and Forest engineering, School of Engineering and Applied Sciences, National University of Mongolia, Ulaanbaatar 210646, Mongolia; Khongor.g20@gmail.com (K.G.); Byambatseren.ch@seas.num.edu.mn (B.C.); Lkhagvajargal@seas.num.edu.mn (L.B.); Ch_sonomdagva@num.edu.mn (S.C.); 3Graduate School of Medical Sciences, Kanazawa University, Kanazawa 920-8640, Japan; jenny830vs1224@gmail.com; 4Institute of Nature and Environmental Technology, Kanazawa University, Kanazawa 920-1192, Japan; nagatogou@se.kanazawa-u.ac.jp (E.G.N.); n_tang@staff.kanazawa-u.ac.jp (N.T.); hayakawa@p.kanazawa-u.ac.jp (K.H.)

**Keywords:** polycyclic aromatic hydrocarbon, total suspended particles, diagnostic ratio, pollution sources, inhalation health risk, Ulaanbaatar, Mongolia

## Abstract

The purpose of this study was to identify pollution sources by characterizing polycyclic aromatic hydrocarbons from total suspended particles in Ulaanbaatar City. Fifteen polycyclic aromatic hydrocarbons were measured in total suspended particle samples collected from different sites, such as the urban center, industrial district and ger (Mongolian traditional house) areas, and residential areas both in heating (January, March), and non-heating (September) periods in 2017. Polycyclic aromatic hydrocarbon concentration ranged between 131 and 773 ng·m^−3^ in winter, 22.2 and 530.6 ng·m^−3^ in spring, and between 1.4 and 54.6 ng·m^−3^ in autumn. Concentrations of specific polycyclic aromatic hydrocarbons such as phenanthrene were higher in the ger area in winter and spring seasons, and the pyrene concentration was dominant in late summer in the residential area. Polycyclic aromatic hydrocarbons concentrations in the ger area were particularly higher than the other sites, especially in winter. Polycyclic aromatic hydrocarbon ratios indicated that vehicle emissions were likely the main source at the city center in the winter time. Mixed contributions from biomass, coal, and petroleum combustion were responsible for the particulate polycyclic aromatic hydrocarbon pollution at other sampling sites during the whole observation period. The lifetime inhalation cancer risk values in the ger area due to winter pollution were estimated to be 1.2 × 10^−5^ and 2.1 × 10^−5^ for child and adult exposures, respectively, which significantly exceed Environmental Protection Agency guidelines.

## 1. Introduction

Ulaanbaatar is the capital city of Mongolia (47°55′13″ N, 106°55′2″ E) which is located at an altitude of about 1350 m above sea level. This city is recognized as the coldest capital city in the world due to geographic features such as high altitude, landlocked location (Figure 1), and a persistent wintertime Siberian High. The area has a cold semi-arid climate according to the Köppen-Geiger climate classification. The monthly average temperature between October and March is below 0 °C, and the average winter temperature is −19 °C [1,2]. The average daily temperature in Ulaanbaatar in winter is about −13 °C, with temperatures dropping to as low as −40 °C at night [3]. To keep residents warm and for cooking, each household in ger (Mongolian traditional house) areas consumes over 5 tons of coal and 3 m^3^ of wood annually during the winter (from November to February) in Ulaanbaatar City [4,5]. The combination of cold temperatures and the geographical setting of the city favors the development of a strong inversion in winter, which traps the smoke emitted from coal and wood combustion. The growing population and increasing demand for energy (heat) has caused the air quality to deteriorate significantly and become the major threat to public health in the city.

Polycyclic aromatic hydrocarbons (PAHs) such as benzo[*a*]pyrene (BaP) are ubiquitous environmental organic pollutants, which mainly originate from imperfect combustion and pyrolysis of organic matter [6,7,8,9,10,11]. Many are known to be carcinogenic and/or mutagenic and have been marked as priority pollutants by both the European Union and the United States Environmental Protection Agency (US EPA). The International Agency for Research on Cancer ranked BaP in Group 1 (carcinogenic to humans), 1-NP in Group 2A (probably carcinogenic to humans), and several other PAHs in Group 2B (possibly carcinogenic to humans). Additionally, PM_2.5_ was categorized as a Group 1 pollutant in 2013, partly because several PAHs and NPAHs, such as BaP and 1-NP can be found in PM_2.5_ [6,7,8,9,12,13,14]. In particular, benzo[*a*]pyrene (BaP) has been selected as an indicator of carcinogenic PAHs [13,14,15] and several countries and organizations have also set up standards for this compound, ranging from 0.1 to 2.5 ng·m^−3^ [16]. The World Health Organization (WHO) air quality guidelines estimate the reference levels of BaP at 0.12 ng·m^−3^ [16]. Several metabolites of PAHs exhibit estrogenic/anti-estrogenic or anti-androgenic activities [8,17,18] or produce reactive oxygen species [8,19,20]. Because PAHs have also been linked to respiratory and cardiovascular diseases, the environmental behaviors of these pollutants need to be more clearly understood [7,8,21].

The main heat sources of Ulaanbaatar City are coal, wood, and used engine oil from vehicles. In addition to the smoke from burning such different types of fuel, the exhaust from diesel powered vehicles is considered a major contributor to the air pollution [6,7]. Although there have been studies on the major chemical composition of ambient particles in the city [22], little is known about the pollution levels of toxic organic compounds such as PAHs when bound to total suspended particles (TSPs) in Ulaanbaatar. Atmospheric PAHs may also cause respiratory problems, impair pulmonary function, and cause bronchitis [12,23]. The objective of this study is to characterize the spatial and temporal variations of PAHs associated with the TSPs in Ulaanbaatar City, identify the sources, and assess the health risks.

## 2. Materials and Methods

### 2.1. Sampling Conditions and Study Area

The TSPs were collected at five locations in Ulaanbaatar City in 2017 at the sampling sites described in Figure 1 and Table 1. Sample point 1 (PAH1) is in an urban center that consists of governmental apartments and educational or administrative institutions situated along wide roads. Sample point 2 (PAH2) is in a ger area consisting of traditional portable houses (also known as yurts) and apartment buildings. Sample point 3 (PAH3) is in a residential area, but also contains some industry, large parking lots, and ger areas. Sample point 4 (PAH4) is located near the wood products industry, a coal distribution center, a gas station, and some ger areas. Sample point 5 (PAH5) represents the southern rural part of the city located near the Tuul River and consists of many modern small townhouses.

### 2.2. Sampling Design and Analysis

#### 2.2.1. Sampling Methods

TSPs were collected on a glass fiber filter (Gb-100R, Toyo Roshi Kaisha, Japan) using a high-volume air sampler (Kimoto Electric Company Limited, Osaka, Japan). The flow rates (300–500 L·min^−1^) and sampling durations (3–24 h) varied depending on the degree of air pollution to avoid filter overloading. The filters were dried overnight in the dark at room temperature. After weighing, the filters were kept in sealed plastic bags and stored at −20 °C until extraction. In total 13 samples were obtained in 2017 (Table 1 and Table 2) by collecting one sample from every sampling site in January, March and September (except for the townhouse area where sampling was performed only in September). Weather conditions such as temperature, precipitation, pressure, wind speed and wind direction during the sampling period are summarized in Table 2.

#### 2.2.2. Sample Pretreatments

An area (diameter 110 mm) of each filter loaded with TSPs was cut into small pieces (ca. 5 × 5 mm^2^). The pieces were placed in a flask and mixed with an aliquot of an ethanol solution containing internal standards (Nap-*d*_8_, Ace-*d*_10_, Phe-*d*_10_, Pyr-*d*_10_ and BaP-*d*_12_) and benzene/ethanol (3:1 v/v). The flask was shaken in an ultrasonic bath to extract PAHs. The extracts were washed successively with diluted sodium hydroxide solution, diluted sulfuric acid solution, and water. After the benzene/ethanol solution was evaporated, the residue was dissolved in acetonitrile [7,8,24,25]. Prior to the evaporation, 100 μL of dimethyl sulfoxide (DMSO) was added for reducing losses of small PAHs which have high vapor pressure. This allowed for the minimal loss of smaller PAHs during the evaporation process. Internal standards such as deuterated Nap (Nap-*d*_8_), Ace (Ace-*d*_10_), Phe (Phe-*d*_10_), Pyr (Pyr-*d*_10_), BaP (BaP-*d*_12_), and DMSO were purchased from Supelco Park (Bellefonte, PA, USA) and Wako Pure Chemicals (Osaka, Japan), respectively. All organic solvents and other reagents used were of special reagent grade [7,8,24,25,26]. Milli-Q purified water (Kanazawa University, Kanazawa, Japan) was also used in this experiment.

#### 2.2.3. Analysis by High-Performance Liquid Chromatography (HPLC)

Aliquots of the solution were then injected into a high-performance liquid chromatography (HPLC) system for the quantification of PAHs. Further details on analytical procedures can be found in previous studies [7,8,24]. In this study we analyzed 15 PAHs with 2–6 rings [8,24,25,26,27]. The 15 PAHs were naphthalene (Nap), acenaphthene (Ace), fluorene (Fle), phenanthrene (Phe), anthracene (Ant), fluoranthene (Flu), pyrene (Pyr), benz[*a*]anthracene (BaA), chrysene (Chr), benzo[*b*]fluoranthene (BbF), benzo[*k*]fluoranthene (BkF), benzo[*a*]pyrene (BaP), dibenz[*a,h*]anthracene (DBA), benzo[*ghi*]perylene (BPe), and indeno[1,2,3-*cd*]pyrene (IDP) [7,8,24,27].

### 2.3. Data Analysis

#### 2.3.1. PAH Source Identification

Different fuel types and combustion temperatures often leave characteristic signatures in the ratios of different PAHs. These ratio profiles can be utilized to identify the emission source. The diagnostic ratio method for identifying PAHs source involves comparison of ratios of frequently found PAHs pairs. PAHs isomer pair ratios such as Ant/(Ant+Phe), Flu/(Flu+Pyr), and BaA/(BaA+Chr) have often been used to distinguish the possible categories of PAHs sources in the environment, due to their relative stability [28,29,30,31,32].

#### 2.3.2. Health Risk Assessment

The risk assessment involves prediction of adverse effects in prolonged exposure to pollution. Potential human carcinogenic risks associated with chemical exposure are expressed in terms of an increased probability of developing cancer during a person’s lifetime. In this work, the risk level of the probability of developing cancer over a lifetime for an adult (15 + 55 = 70 years) and children (15 years) [33], respectively, was determined. This risk level was estimated by multiplying the slope factor (SF) by the life average daily dose for the carcinogenic substance (LADD); in turn, LADD is obtained by multiplying the concentration of a substance (CC) by the intake factor (IF). The equation used for the calculation of the risk level is as follows [30,34,35].

RISK = LADD × SF(1) where LADD is life average daily dose for carcinogenic substance coinciding with chronic daily intake (CDI), expressed as:CDI (LADD) = CC × IF(2) where CC is concentration of each compound (mg·m^−3^) and IF is an intake factor (m^3^ kg^−1^ day^−1^).

Intake factor is derived from Equation (3):IF = IR × ED × EF × ET/(BW × AT)(3) where IR is the inhalation rate, corresponding to the breathing rate (m^3^·day^−1^, 20 for adults and 7.6 for children); ED is the lifetime exposure duration (55 and 15 years for adult and children, respectively); EF is the exposure frequency (120 days·year^−1^, by accounting for the heating season from November to February); ET is the exposure time (24 h·day^−1^); BW is the body weight (70 kg and 15 kg for adults and children); and AT is the average time of an average exposure extent over a lifetime (70 and 15 years for adult and children, respectively).

A Slope Factor (mg·kg^−1^·day^−1^) is an estimate of the probability of the response per unit chemical intake over a lifetime. It is used to estimate the probability of an individual developing cancer as a result of the lifetime exposure to a certain level of potential carcinogen [34,35]. The SF depends on the inhalation unit risk (IUR) and is the potency factor for inhalation exposure (Table 3). The SF values were calculated by Equation (4):SF = IUR × 1000 × BW/ IR(4) where 1000 is the conversion factor (mg·µg^−1^).

The human health risk related to contaminated air depends on the extent of exposure as well as on the toxic effects of chemicals. A final health risk level is expressed as the sum of the individual risks of each compound. A significant total risk is found when the risk exceeds 1 × 10^−3^ [32,35].

These health risk values indicated that the daily inhalation dose of PAHs and cancer risk to adults and children residing around the sampling areas were comparable in the heating (January and March months) and non-heating (September) periods to the acceptable levels of 10^−6^ to 10^−4^ as proposed by the U.S.EPA [16,36,37].

## 3. Results and Discussion

### 3.1. Meteorological Condtions and Seasonal Variation

The surface air temperatures first dropped below 0 °C in October, which prompted households in the ger (traditional Mongolian dwelling) districts to start using heating. The maximum atmospheric boundary layer height continuously decreased from summer to winter. Stable atmospheric conditions and a surface inversion layer in the winter resulted in low wind velocities (<2 m·s^−1^), especially at night as reported by the previous study [38]. Consequently, because of both the meteorological and topographical conditions, air pollutants remained stagnant at the urban surface level, which resulted in high concentrations of PM_2.5_ in the winter as shown in Table 2.

Maximum wind speed was strongest on 14 September (14 m·s^−1^) and weakest on 21 January (4 m·s^−1^) during the sampling periods. The maximum temperature in the urban area was 23 °C on 12 September and the minimum was −28 °C on 17 January. The diurnal variation in the temperature is depicted in Table 2. Generally, wind speed is stronger in the warm period than in the cold period, and precipitation and relative humidity are lower in March and September than in January.

### 3.2. Characteristics of Atmospheric PAHs Concentrations in Ulaanbaatar City

The changes in total atmospheric PAH concentrations (2-, 3-, 4-, 5- and 6-ring PAHs) in Ulaanbaatar City are shown in Figure 2, Figure 3 and Table 4.

The concentrations of total PAHs at these sites were the highest in ger areas (reaching 773 ng·m^−3^ in January), followed in descending order by residential, industrial, city center, and townhouse areas (Figure 2). The spatial variations most likely reflect the proximity, type and source strength of the emissions near the sampling sites. These seasonal changes basically follow the greater heating demand in colder months. Due to limited industrial activity and modern construction, lower levels of emission are expected in the townhouse area than other sampling sites in Ulaanbaatar. Indeed, the concentration of total PAHs in the townhouse area was 1.4 ng·m^−3^ in the late summer period, which was the lowest value recorded in all sampling sites. The average concentration of total PAHs during the late summer period was 23 and 15 times lower than those for winter and spring, respectively.

The ratio of 3-ring PAHs was higher in January and March than in September (Figure 3). Phenanthrene (Phe) was dominant in the ger area in the winter. PAHs can exist in both particulate and gas phases depending on their volatility. Generally, PAHs with smaller ring numbers (≤3) such as Ace, Phe, Flu, and Ant, tend to be more volatile than those with larger ring numbers (≥4) (which exist preferentially in the particulate phase). Partitioning depends on temperature; colder temperatures cause greater partitioning into the particulate phase. Therefore, the larger fractions of 4-ring PAHs shown in Figure 3 may not only reflect the source profile but may also be a result of the low temperature conditions encountered in the coldest capital, especially in the heating seasons.

The PAH concentrations in this study were compared with the PAHs concentrations measured in other cities. Table 5 and Figure 4 shows PAHs concentrations during the heating (H) and non-heating (NH) periods. During the heating (January) and non-heating periods in Ulaanbaatar, maximum PAHs concentrations were 773 ng·m^−3^ and 53.1 ng·m^−3^. The maximum concentration of PAHs during the winter period was 1.2, 5.5, 20.3, 618.4, and 533.1 times higher than those of Tangshan, Beijing, Seoul, Kanazawa, and Tokyo, respectively.

Table 6 shows the correlations between the meteorological variables and PAHs concentrations presented in Table 2 and Table 4, except for wind direction in the city center. The most important negative correlations were expressed between temperature and both total PAHs concentration (*r* = −0.97) and PM_2.5_ concentration (*r* = −0.95). A negative correlation was also indicated between wind speed and total PAHs concentration (*r* = −0.98). A significant correlation was observed between PM_2.5_ concentrations and total PAH concentration (*r* = 0.85). As expected, low temperatures during cold periods facilitate the development of an inversion layer in the valley. Stagnant air then traps the considerable heating emissions. The negative correlation shown in Table 6 adequately illustrates the dilemma of a city where residents are forced to choose heat over air quality.

### 3.3. Composition and Primary PAH Sources

Diagnostic ratios have been applied to identify the possible sources of PAHs [22,26,27,28,32]. Table 7 lists the commonly used diagnostic ratios. For example, Flu/(Flu+Pyr) ratios lower than 0.40 are characteristic of petroleum sources, while values between 0.4 and 0.5 are characteristic of petroleum combustion. If the ratio exceeds 0.5, the source is regarded as biomass and coal combustion [30,40,41]. BaA/(BaA+Chr) ratios ranging between 0.38–0.64 are considered as good markers for diesel engines and 0.22–0.55 for gasoline emissions [30,41]. For Ant/(Ant+Phe) ratios, 0.1 is taken as a threshold to distinguish petroleum from combustion sources [30,41].

In this study, we applied the PAHs pair ratios Ant/(Ant+Phe), BaA/(BaA+Chr) and Flu/(Flu+Pyr) as producers of the sources of PAHs in heating and non-heating periods in the sampling sites.

The PAH ratios (Figure 5) show that Flu/(Flu+Pyr) ratio ranged from 0.2 to 0.88, indicating significant contribution of biomass and coal combustion, especially in ger area during the heating period. There is a general trend for a stronger influence from coal and wood combustion in the colder period (winter), while dominance from petroleum sources are expressed in the townhouse and industrial areas in the late summer (September). Plots from the spring (March) are found in the intermediate range between the height of the heating and non-heating seasons, showing mixed characteristics.

The scatter plot (Figure 6) showed that BaA/(BaA+Chr) varied between 0.44 and 0.66, suggesting stronger contribution of biomass and coal combustion, especially during the winter sampling period. For Ant/(Ant+Phe), the ratios ranged from 0.01 to 0.2, indicating a strong influence from petroleum in the city center and residential area, and biomass and coal combustion in the ger and industrial areas. There is a general trend that in colder periods (winter and spring), the plots tend to shift towards toward higher ratios, indicating a stronger influence from coal and wood combustion. This is particularly pronounced in the ger area. Meanwhile, the ratios for the city center and town houses tended to be lower, indicating the stronger influence from vehicle emissions irrespective of the seasons.

In summary, vehicle emission was suggested as the main source of PAHs at the city center even during winter. In other sampling sites, mixed contribution from the biomass and coal combustion, as well as vehicle emission (petroleum combustion) were suggested, but the biomass and coal combustion were prominent in colder periods (winter and spring) especially in the ger, industrial and residential areas. This demonstrates a direct link between high PAH volumes observed in respective areas (Figure 2) with the fuel that is being consumed for heating purposes in colder periods.

### 3.4. Health Risk Assessment

Inhalation cancer risk level expressed as the sum of the individual risks from each compound. Lifetime cancer risks of adult and children are shown in Figure 7. In the ger area during the heating period, inhalation cancer risk values were 1.2 × 10^−5^ and 2.1 × 10^−5^ for child and adult exposure, respectively. The inhalation cancer risk levels of adults and children were 1.5 times higher in the ger area than in the city center area in the heating periods. BaP is the most important contributor to the inhalation cancer risk and in the ger area levels were 22 ng·m^−3^ and 1.4 ng·m^−3^ in the heating and non-heating periods, respectively. Figure 7 show the risk level for each compound and demonstrates that the total PAH risk level was higher than EPA critical value. It also suggested that the cancer risk to humans was substantially higher in the cold seasons and that drastic temporal control measures are needed to effectively mitigate the health risks.

## 4. Conclusions

The concentrations of PAHs in ambient TSP in Ulaanbaatar City were measured in heating (January and March) and non-heating (September) periods in 2017. To the best of our knowledge, this is the first study to report on the seasonal as well as spatial variations of particulate PAHs pollution in the city. The total PAHs concentrations ranged between 131 and 773 ng·m^−3^ in the winter, 22.2 and 530.6 ng·m^−3^ in spring and 1.4 and 54.6 ng·m^−3^ in autumn. Concentrations of PAHs in the ger area was considerably higher than other sampling sites, especially in the winter.

The ratio of PAHs indicated that vehicle emissions were the major source of PAHs in the city center even in the winter time. At other sampling sites, a mixed contribution from the biomass and coal combustion, as well as petroleum combustion were suggested during the whole observation period. However, wood and coal combustion gained relative importance in colder months especially in ger and residential areas due to the high heating demand. On the other hand, petroleum combustion was the major source of PAHs during the warmer seasons in Ulaanbaatar City. This study has shown that the lifetime cancer risks to be substantially higher with prolonged exposure to winter air pollution, which highlights the urgent need in the city to implement drastic mitigation measures.

## Figures and Tables

**Figure 1 ijerph-16-00442-f001:**
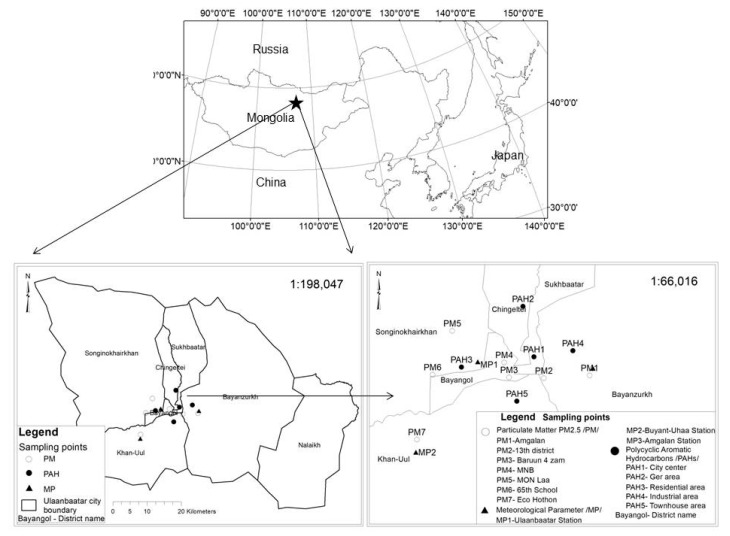
Sampling sites in Ulaanbaatar City, Mongolia. Boundaries in the lower panels show municipal districts (*düürgüüd*) within Ulaanbaatar City.

**Figure 2 ijerph-16-00442-f002:**
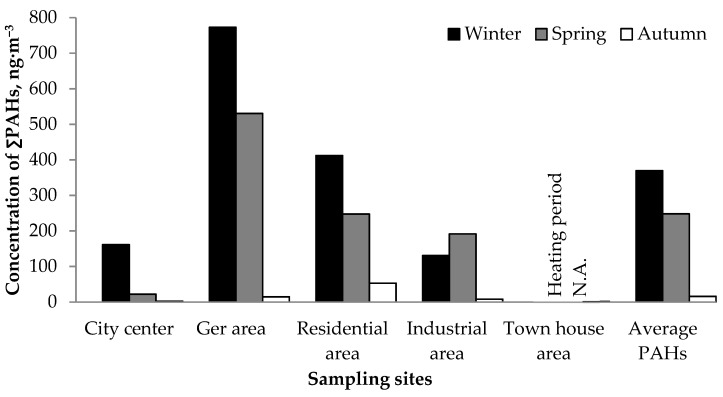
Atmospheric concentrations of PAHs in Ulaanbaatar City.

**Figure 3 ijerph-16-00442-f003:**
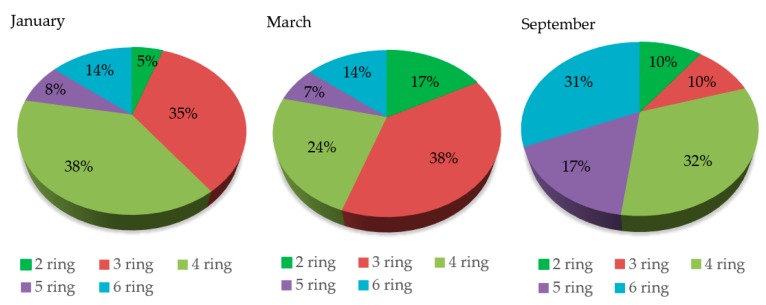
Fractions of PAHs with different ring numbers.

**Figure 4 ijerph-16-00442-f004:**
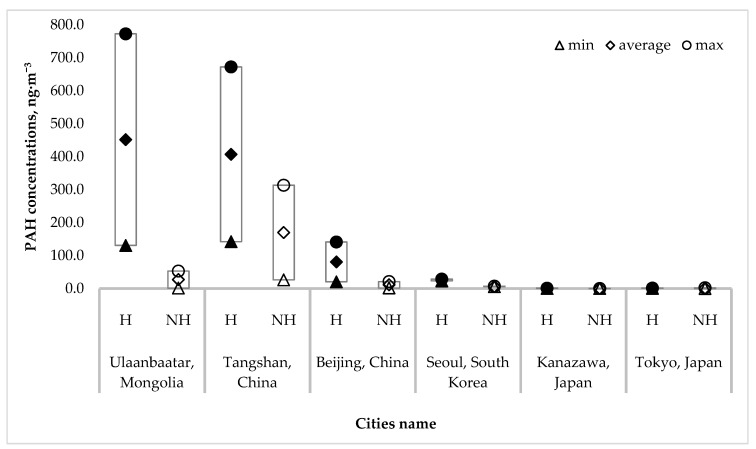
Compared concentrations of PAHs in the heating (H) and non-heating (NH) periods of select Asian cities.

**Figure 5 ijerph-16-00442-f005:**
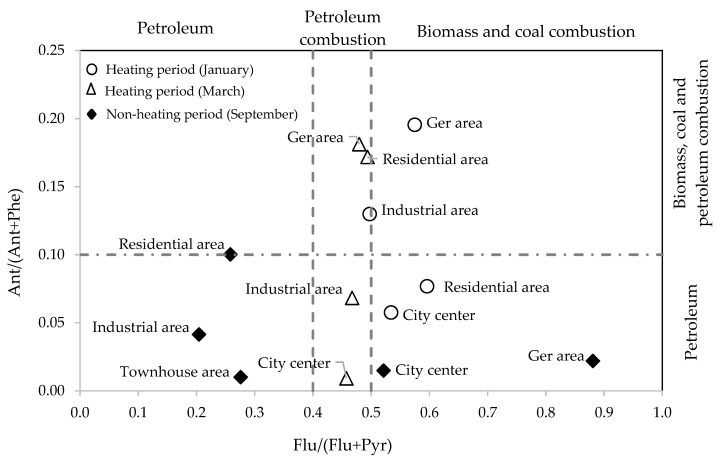
Cross plot of the diagnostic ratios for the sources of PAHs in winter and late summer.

**Figure 6 ijerph-16-00442-f006:**
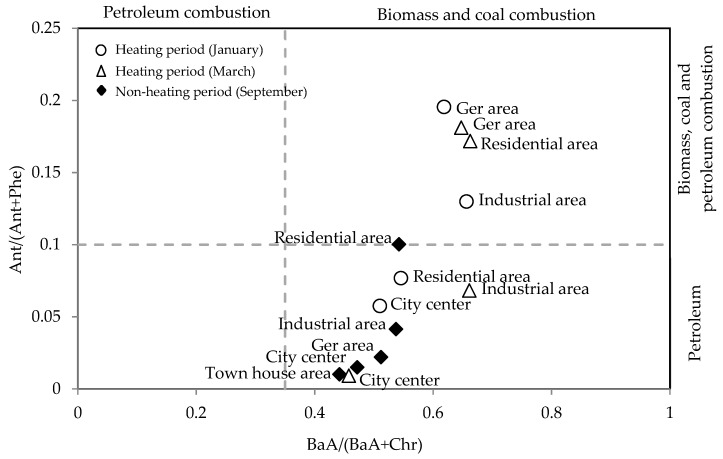
Cross plot of the diagnostic ratios for the sources of PAHs in winter and late summer.

**Figure 7 ijerph-16-00442-f007:**
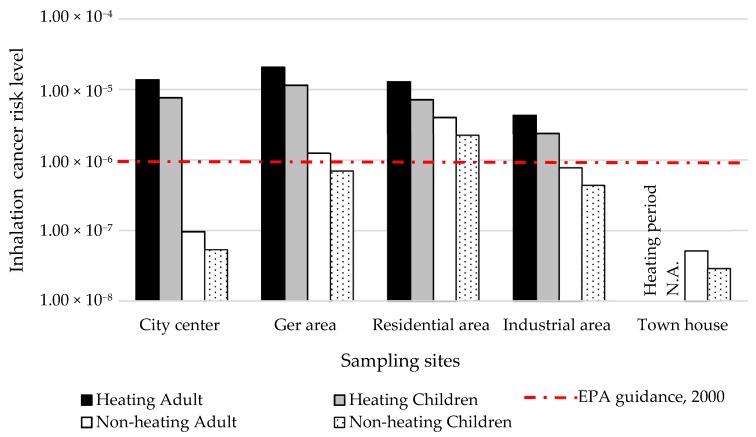
Estimated lifetime inhalation cancer risk attributed to measured concentration of PAHs in the ambient air of different sampling sites of Ulaanbaatar City.

**Table 1 ijerph-16-00442-t001:** Sampling station locations in the Ulaanbaatar City, Mongolia.

Parameters	Sample Code	Sampling Station Name and Types	Station Coordinates	
Polycyclic aromatic hydrocarbons (PAHs)	PAH1	City center	47°55′21.8″ N	106°55′13.5″ E
	PAH2	Ger area	47°58′1.2″ N	106°54′28.8″ E
	PAH3	Residential area	47°54′51.9″ N	106°49′37.3″ E
	PAH4	Industrial area	47°55′34.6″ N	106°58′18.7″ E
	PAH5	Townhouse area	47°53′2.5″ N	106°53′49.9″ E
Particulate matter	PM1	Amgalan (industrial area)	47°54′19.1″ N	106°59′32.8″ E
	PM2	13th district (city center area)	47°54′14.2″ N	106°55′59.1″ E
	PM3	Baruun 4 zam (city center area)	47°54′19.0″ N	106°53′16.7″ E
	PM4	MNB (ger area)	47°55′6.5″ N	106°52′56.2″ E
	PM5	Mon laa (ger area)	47°56′48.9″ N	106°48′57.5″E
	PM6	65th school (residential area)	47°54′33.3″ N	106°47′20.3″ E
	PM7	Eco khotkhon (apartment)	47°51′10.0″ N	106°46′0.2″ E
Meteorological Parameters	MP1	Ulaanbaatar station	47 55′8.6″ N	106 50′51.9″ E
	MP2	Buyant Ukhaa station	47 50′29.4″ N	106 45′52.9″ E
	MP3	Amgalan station	47 54′40.0″ N	106 59′5.0″ E

**Table 2 ijerph-16-00442-t002:** Meteorological conditions and PM_2.5_ and PAHs concentrations during the sampling period.

Sampling Sites	Day, Month, Year	Temperature, °C			Wind Speed, m·s^−1^		Wind Direction		Relative Humidity, %	Precipitation, mm	Pressure, hPa	PM_2.5_ Concentration, µg·m^−3^	PAH Concentration, ng·m^−3^
		Max	Min	Mean	Max	mean	Direction	Angle					
City center	17, 01, 2017	−21.2	−28	−24.9	5	0.8	-	0	73	0.3	872.9	172 ^a^	161.6
Ger area	21, 01, 2017	−16.9	−27.1	−23.1	4	0.9	-	0	74	0	875.9	252 ^b^	773.0
Residential area	22, 01, 2017	−16.8	−27.4	−22.6	6	1	EEN	84	67	0	874.7	235 ^c^	412.3
Industrial area	24, 01, 2017	−8.9	−21.5	−16.3	6	1.9	EES	113	65	0	873.1	68 ^d^	131.0
City center	15, 03, 2017	4.6	−9.8	−3.2	7	1.4	E	101	45	0	871.9	54.5 ^a^	22.2
Ger area	16, 03, 2017	6.4	−7	−0.7	7	1.3	EES	113	36	0	866.5	87 ^b^	530.6
Residential area	19, 03, 2017	0.9	−11.9	−5.8	9	1.5	-	0	53	0	876.7	27 ^c^	247.5
Industrial area	20, 03, 2017	4.6	−10.7	−3.8	8	1.4	-	0	42	0	876.3	22 ^d^	191.4
City center	12, 09, 2017	23	6	14	7	2	ES	135	50	0	871.0	19 ^a^	2.2
Ger area	14, 09, 2017	21.7	12	16.5	14	3.5	NEN	17	44	0	868.6	20 ^b^	14.4
Residential area	19, 09, 2017	21.2	1.5	10.9	9	1.6	SWS	208	48	0	869.2	57 ^c^	53.1
Industrial area	21, 09, 2017	17.5	1.4	6.2	13	4.1	W	343	72	0	856.4	13 ^d^	7.8
Town house	23, 09, 2017	12.1	5.9	7.3	11	1.9	W	343	74	0	860.2	8 ^e^	1.4

^a^ Average UB2 and UB4 stations; ^b^ Zuragt station; ^c^ Tolgoit station; ^d^ Amgalan station; ^e^ Nisekh station., N-north, W-west, S-south, E-east, ES-east-south, EEN-east, east-north, EES-east, east-south, NEN-north, east-north, SWS-south, west-south.

**Table 3 ijerph-16-00442-t003:** Inhalation unit risk (IUR) for the studied PAHs.

PAHs Species	Abbreviation	Chemical Formula	MW, g/mol	Rings	MW Groups	IUR, (µg m^−3^)^−1 ᵃ^
Naphthalene	Nap	C_10_H_8_	128.2	2	LMW	3.4 × 10^−5^
Acenaphthene	Ace	C_12_H_10_	154.2	3	LMW	1.1 × 10^−6^
Fluorene	Fle	C_13_H_10_	166.2	3	LMW	1.1 × 10^−6^
Phenanthrene	Phe	C_14_H_10_	178.2	3	LMW	1.1 × 10^−6^
Anthracene	Ant	C_14_H_10_	178.2	3	LMW	1.1 × 10^−5^
Fluoranthene	Flu	C_16_H_10_	202.3	4	MMW	1.1 × 10^−6^
Pyrene	Pyr	C_16_H_10_	202.3	4	MMW	1.1 × 10^−6^
Benz[*a*]anthracene	BaA	C_18_H_12_	228.3	4	MMW	1.1 × 10^−4^
Chrysene	Chr	C_18_H_12_	228.3	4	MMW	1.1 × 10^−5^
Benzo[*b*]fluoranthene	BbF	C_20_H_12_	252.3	5	HMW	1.1 × 10^−4^
Benzo[*k*]fluoranthene	BkF	C_20_H_12_	252.3	5	HMW	1.1 × 10^−4^
Benzo[*a*]pyrene	BaP	C_20_H_12_	252.3	5	HMW	1.1 × 10^−3^
Dibenz[*a,h*]anthracene	DBA	C_22_H_14_	278.4	5	HMW	1.2 × 10^−3^
Benzo[*ghi*]perylene	BPe	C_22_H_12_	276.3	6	HMW	1.1 × 10^−5^
Indeno[1,2,3*-cd*] pyrene	IDP	C_22_H_12_	276.3	6	HMW	1.1 × 10^−4^

MW: molecular weight; ^ᵃ^ Silvia et al. (2014) [35].

**Table 4 ijerph-16-00442-t004:** The selected PAH species, concentrations, and groupings by molecular weight in the urban ambient air of Ulaanbaatar.

Sampling Sites	Sampling Period	PAHs Concentrations, ng·m^−3^															
		Nap	Ace	Fle	Phe	Ant	Flu	Pyr	BaA	Chr	BbF	BkF	BaP	DBA	BPe	IDP	ΣPAHs
City center	17, 01, 2017	5.8	0.05	4.1	30.7	1.9	35.1	30.6	6	5.8	5.6	2.8	3.6	14.2	12.8	2.4	161.6
Ger area	21, 01, 2017	5.3	1.8	134.9	175.5	42.6	143.5	106.1	30.3	18.7	26.2	9.9	22	0.5	44.8	11	773
Residential area	22, 01, 2017	5.2	0.2	21.9	109.4	9.1	99.4	67.4	16.5	13.7	15.1	6.1	13.8	0.5	25.6	8.5	412.3
Industrial area	24, 01, 2017	5.6	0.1	6.2	31.1	4.6	23.5	23.7	7.7	4	5.7	2	4.4	0	9.9	2.3	131
City center	15, 03, 2017	3.2	0.04	0.4	11.6	0.1	1	1.2	0.3	0.4	0.8	0.3	0.5	0.1	1.4	0.7	22.2
Ger area	16, 03, 2017	19.4	1.8	38.9	177.9	39.4	53.8	58.4	20.2	11	24.4	9.5	21.2	0.5	43.3	10.9	530.6
Residential area	19, 03, 2017	17	0.1	20.5	47	9.8	46	47.2	17	8.7	8.2	2.8	6.2	0.3	14.2	2.6	247.5
Industrial area	20, 03, 2017	13.4	2.3	104.2	17.5	1.3	10.8	12.3	2.4	1.2	6.4	1.3	3.5	0.3	14	0.4	191.4
City center	12, 09, 2017	0.3	0.003	0.04	0.9	0.01	0.2	0.2	0.1	0.1	0.1	0.04	0.1	0.04	0.2	0.1	2.2
Ger area	14, 09, 2017	0.6	0.02	0.5	1	0.02	2.5	0.3	1.2	1.1	1.6	0.7	1.4	0.02	2.5	0.9	14.4
Residential area	19, 09, 2017	0.8	0.04	1.8	4	0.4	3	8.6	7.6	6.4	4.9	2	4.3	0.01	7.9	2.7	54.6
Industrial area	21, 09, 2017	0.6	0.02	0.4	0.8	0.03	0.2	0.6	0.6	0.5	0.9	0.3	0.9	0.0003	1.3	0.8	7.8
Town house	23, 09, 2017	0.4	0.001	0.03	0.6	0.01	0.04	0.1	0.03	0.04	0.03	0.03	0.1	0.001	0.1	0.002	1.4
IUR		3.4 × 10^−5^	1.1 × 10^−6^	1.1 × 10^−6^	1.1 × 10^−6^	1.1 × 10^−5^	1.1 × 10^−6^	1.1 × 10^−6^	1.1 × 10^−4^	1.1 × 10^−5^	1.1 × 10^−4^	1.1 × 10^−4^	1.1 × 10^−3^	1.2 × 10^−3^	1.1 × 10^−5^	1.1 × 10^−4^	

ΣPAHs = Nap + Ace + Fle + Phe + Ant + Flu + Pyr + BaA + Chr + BbF + BkF + BaP + DBA + BPe + IDP.

**Table 5 ijerph-16-00442-t005:** The temperature and PAH concentrations in other Asian cities. TSPs: total suspended particles.

No.	City, Country	Type	Temperature, °C		Fraction	ΣPAHs Concentrations Range, ng·m^−3^		Author(s)
			Summer	Winter		Summer	Winter	
1	Tangshan, China	Industrial and Commercial	27.3	−2.7	PM_10_	26.5–313.6	142.4–672.4	Shi et al. (2009) [36]
2	Beijing, China	Commercial	23.7	1.5	PM_2.5_	1.8–21.2	20.7–141.3	Wu et al. (2014) [39]
3	Seoul, South Korea	Commercial	24.5	−3.4	PM_10_	5.8–7.2	23.4–28.8	Kim et al. (2012) [37]
4	Kanazawa, Japan	Commercial	25.1	3.7	TSPs	0.28–0.44	0.75–1.25	Hayakawa et al. (2018) [16]
5	Tokyo, Japan	Commercial	25.9	6.6	TSPs	0.12–0.24	0.95–1.45	
6	Ulaanbaatar, Mongolia	Commercial and Industrial	14	−24.9	TSPs	1.4–53.1	131.0–773.0	This study

**Table 6 ijerph-16-00442-t006:** Correlations between PM_2.5_ and PAH concentrations (for total and different rings numbers), and meteorological parameters.

Parameters	ΣPAHs	PM_2.5_	Temp	WS	RH	Pre	Press	2-ring	3-ring	4-ring	5-ring	6-ring
ΣPAHs	1											
PM_2.5_	0.85	1										
Temp	−0.97	−0.95	1									
WS	−0.98	−0.74	0.91	1								
RH	0.31	0.77	−0.53	−0.13	1							
Pre	0.76	0.99	−0.90	−0.63	0.85	1						
Press	0.98	0.72	−0.90	−1.00	0.10	0.61	1					
2-ring	0.54	0.01	−0.32	−0.69	−0.63	−0.13	0.71	1				
3-ring	0.98	0.74	−0.91	−1.00	0.14	0.64	1.00	0.68	1			
4-ring	0.97	0.95	−1.00	−0.91	0.53	0.90	0.90	0.32	0.91	1		
5-ring	1.00	0.87	−0.98	−0.97	0.36	0.80	0.97	0.50	0.97	0.98	1	
6-ring	1.00	0.81	−0.95	−0.99	0.25	0.72	0.99	0.60	0.99	0.95	0.99	1

**Σ**PAHs: total PAH concentrations, ng·m^−3^, PM_2.5_: PM_2.5_ concentrations, µg·m^−3^, Temp: temperature, °C, WS: wind speed, m·s^−1^, RH: relative humidity, %, Pre: precipitation, mm, Press: pressure, hPa.

**Table 7 ijerph-16-00442-t007:** Diagnostic ratios of PAHs for samples from previous studies.

Diagnostic Ratio (DR)	Ant/(Ant+Phe)	Flu/(Flu+Pyr)	BaA/(BaA+Chr)
Petroleum	<0.1 [30] ^a^	<0.4 [30] ^a^	-
Gasoline engine	-	0.4–0.5 [30] ^a^	0.22–0.55 [42] ^b^
Diesel engine	-	>0.5 [43] ^c^	0.38–0.64 [42] ^b^
Coal combustion	0.24 [15] ^d^	0.57 [15] ^a^	0.5–0.55 [15] ^e^
Wood combustion	-	-	0.43 [44] ^f^

ᵃ Li et al. (2016); ᵇ Sicre et al. (1987); ^c^ Ravindra et al. (2008); ᵈ Kong et al. (2010); ^e^ Shi et al. (2014); ^f^ Li et al. (1993).

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
