# Peer review of "Sources and Characteristics of Polycyclic Aromatic Hydrocarbons in Ambient Total Suspended Particles in Ulaanbaatar City, Mongolia"

_ijerph, 2019, doi:10.3390/ijerph16030442_

Round 1
Reviewer 1 Report
Correct orthographical variants ("PAHs" and "PAH”). This can confuse readers.
Insert space between ng and m^-3.
Table 2. Chemical formula of DBA is not correct.
L178. Authors should mention result presented in Fig.2 and Fig. 3 in the text.
L212. Use Italic for “r”.
Author Response
Dear Reviewer 1
We changed minor revisions following your comments.
Thank you so much.
Reviewer 2 Report
Byambaa et al. quantified PAH in ambient TSP in Ulaanbaatar, Mongolia. While I have some concerns about the quality of their measurements and this paper does not bring major technical advancement or major findings in terms of new science, its quantification and source identification of the PAH pollution in Ulaanbaatar, which is among the most polluted cities in the world and whose air pollution urgently needs investigations, certainly provide enough merits. Also this paper is generally well written. Therefore, I recommend the publication of this paper in IJERPH after the following issues are addressed.
In the paper, the authors made a number of tables containing a large amount of data. I understand that numbers themselves are of interest, especially for Ulaanbaatar. It would be much better if these data are presented in a more graphical manner (e.g., Figs. 5 and 6).
Line 94: I am wondering how to tell if filter is not clogged
Lines 102-106: This procedure has several solvent changes and an evaporation process. Sample loss in this procedure should be quantified, especially for the solvent evaporation process, as smaller PAHs may also evaporate significantly.
Table 2: Are IUR data obtained in this study or elsewhere? Please provide relevant references if they are not results of this study. Otherwise, please describe how they are estimated.
Lines 166 and 169: the numbers of the lowest wind speed are contradictory in these 2 lines: <2 m/s in Line 166 and 4 m/s in Line 169.
Table 3: Similar issue as above. Some important data shown on a map would help a lot.
Figure 4: I doubt that max and min values are really meaningful. They should be very sensitive to physical and meteorological conditions. What about using, for example, 95% confidence interval instead?
Technical corrections:
Line 52: The European Community no longer exists. Do the authors mean the European Union or the European Commission here?
Figure 1: Are the boundaries in this figure district boundaries within Ulaanbaatar City or something else? Please explain.
Line 166: “ms-1” -> “m s-1”
Author Response
Responses to Reviewer 2 Comments
Point 1: In the paper, the authors made a number of tables containing a large amount of data. I understand that numbers themselves are of interest, especially for Ulaanbaatar. It would be much better if these data are presented in a more graphical manner (e.g., Figs. 5 and 6).
Response 1: We agree that some tables in the manuscript carry a large amount of data. We know for example that Table 5 basically tells the same information as figure 4, but it is also difficult for the readers to extract exact values (e.g. for Korea and Japan) only from the figure. So, we prefer to keep both the table and the figure as they are in this case.
Point 2: Line 94: I am wondering how to tell if filter is not clogged
Response 2: As pointed out by the reviewer, it is indeed difficult to tell if the filter is clogged. We have changed the expression from “clogging” to “overloading” to avoid confusion.
Point 3: Lines 102-106: This procedure has several solvent changes and an evaporation process. Sample loss in this procedure should be quantified, especially for the solvent evaporation process, as smaller PAHs may also evaporate significantly.
Response 3: Prior to the evaporation, 100 ul of dimethyl sulfoxide (DMSO) was added for reducing losses of small PAHs which have high vapor pressure. This allowed for the minimal loss of smaller PAHs during the evaporation process. We added about it in the text.
Point 4: Table 2: Are IUR data obtained in this study or elsewhere? Please provide relevant references if they are not results of this study. Otherwise, please describe how they are estimated.
Response 4: We referred to other work for the IUR data. The reference is now added in the table.
Point 5: Lines 166 and 169: the numbers of the lowest wind speed are contradictory in these 2 lines: <2 m/s in Line 166 and 4 m/s in Line 169.
Response 5: The former value (<2 m s-¹) is based on a previous study explaining about more general condition and the latter value (4 m s-¹) is what was observed during the sampling period. This has been clarified in the revised text.
Point 6: Table 3: Similar issue as above. Some important data shown on a map would help a lot.
Response 6: We agree that some tables in the manuscript carry a large amount of data. In Table 3 the only information used is with regards to the PAHs concentration and so the rests are into the Supplementary Information.
Point 7: I doubt that max and min values are really meaningful. They should be very sensitive to physical and meteorological conditions. What about using, for example, 95% confidence interval instead?
Response 7: We agree that the PAHs concentrations depend largely on physical and meteorological conditions. However, our data is not based on continuous sampling at a fixed location. Instead, we showed few cases from different geographical locations in the city. In this case, we may not have statistically sound parameter to show confidence intervals, and our result should rather be seen as case studies showing the range of concentrations that can be expected depending on the location within the city. Therefore, we believe we do not have much choice but to keep the current expression.
Technical corrections:
Point 8: Line 52: The European Community no longer exists. Do the authors mean the European Union or the European Commission here?
Response 8: It is the European Union. I have corrected the text.
Point 9: Figure 1: Are the boundaries in this figure district boundaries within Ulaanbaatar City or something else? Please explain.
Response 9: The boundaries in the figure show municipal districts (düürgüüd) within Ulaanbaatar City. This is now explained in the figure caption.
Point 10: Line 166: “ms-1” -> “m s-1”
Response 10: Space has been inserted between m and s-1.
